# The Genotyping Diversity and Hemolytic Activity of *Cronobacter* spp. Isolated from Plant-Based Food Products in Poland

**DOI:** 10.3390/foods12203873

**Published:** 2023-10-23

**Authors:** Monika Garbowska, Anna Berthold-Pluta, Lidia Stasiak-Różańska, Antoni Pluta, Stephen Forsythe, Ilona Stefańska

**Affiliations:** 1Department of Technology and Food Assessment, Division of Milk Technology, Institute of Food Sciences, Warsaw University of Life Sciences—SGGW, 02-787 Warsaw, Poland; anna_berthold@sggw.edu.pl (A.B.-P.); lidia_stasiak_rozanska@sggw.edu.pl (L.S.-R.); antoni_pluta@sggw.edu.pl (A.P.); 2FoodMicrobe.com, Keyworth, Nottinghamshire NG12 5GY, UK; sforsythe4j@gmail.com; 3Department of Preclinical Sciences, Institute of Veterinary Medicine, Warsaw University of Life Sciences—SGGW, 02-787 Warsaw, Poland; ilona_stefanska@sggw.edu.pl

**Keywords:** *Cronobacter* spp., MLST, hemolysis, sequence types, clonal complexes, sprouts, nuts, ready-to-eat leaf vegetables

## Abstract

The present study aimed to determine the genotyping diversity and hemolytic properties of 24 strains of *Cronobacter* spp. (15 *Cronobacter sakazakii*, 6 *Cronobacter malonaticus*, 2 *Cronobacter turicensis*, and 1 *Cronobacter condimenti*) isolated from commercial ready-to-eat leaf vegetables, sprouts, nuts, and dried fruits. The multilocus sequence typing (MLST) method was used to determine the sequence types (ST) and clonal complexes (CC) of these strains. The study demonstrated the high genotypic diversity of the *Cronobacter* genus bacteria isolated from plant-based foods. Five novel sequence types (804, 805, 806, 807, and 808) and the presence of novel alleles in the *ppsA*, *gltB*, *gyrB*, and *infB* loci were detected. In total, 16 of the 24 strains were assigned to the sequence types ST99, ST258, ST17, ST648, ST21, ST494, and ST98. One *C. sakazakii* strain (s12) isolated from alfalfa sprouts was assigned to the clonal complex CC4, which encompasses strains often associated with severe infections leading to meningitis in infants. In addition, 87.5% and 16.7% of the *Cronobacter* spp. strains showed β-hemolysis of equine and sheep red blood cells, respectively. The presence of the pathogenic species *C. sakazakii*, *C. malonaticus*, and *C. turicensis* in ready-to-eat plant-derived food products shows they are potential sources of infection, especially to those with compromised immunity, which substantiates their further multi-faceted characterization. The significance of this study may prove useful not only in epidemiological investigations, but also in assessing the risk of infections caused by the presence of *Cronobacter*.

## 1. Introduction

*Cronobacter* bacteria are motile, facultatively anaerobic, Gram-negative rods of the family *Enterobacteriaceae*. The history of this genus is relatively short, as it was only in 2007 that organisms previously classified on the basis of their phenotypic features as *Enterobacter sakazakii* (and even earlier as yellow-pigment-producing *Enterobacter cloacae*) were assigned to the new genus *Cronobacter*, then composed of four species. However, as knowledge of the diversity of the genus improved, it was expanded to the currently agreed composition of seven species. Of the seven *Cronobacter* species, *C. sakazakii, C. malonaticus*, and *C. turicensis* have been recognized as important human pathogens resulting in opportunistic infections [1,2]. These bacteria can cause severe, invasive infections in preterm infants, low-birth-weight infants, neonates hospitalized in intensive care, and immunocompromised infants, as well as immunocompromised adults and the elderly [2,3]. Infections caused by *Cronobacter* spp. in infants are associated with a high mortality rate, ranging from 40 to 80% [4,5], with frequent neurological complications observed in 20% of convalescents [6,7]. *Cronobacter* bacteria cause bacteremia, necrotizing enterocolitis, meningitis, and sepsis in neonates and infants, and wound and urinary tract infections in adults [7,8]. In 2002, the International Commission for Microbiological Specification for Foods (ICMSF) placed *Cronobacter* (then known as *Enterobacter sakazakii*) in group I.B., which covers pathogenic bacteria that are a “severe hazard for restricted populations, life threatening or substantial chronic sequelae or long duration” [9]. The number of *Cronobacter* infections reported in the literature is relatively low and is most likely underestimated, as it is generally not a mandated notifiable disease [7,10].

The prevalence of *Cronobacter* spp. in various food products has been confirmed in investigations conducted across many countries [11,12,13,14,15,16,17,18,19,20,21,22,23,24,25,26], with *C. sakazakii* being the most frequently isolated species. The presence of pathogenic *C. sakazakii*, *C. malonaticus*, and *C. turicensis* in ready-to-eat food products (RTE) makes them putative sources of infections [16,19,20,21,22,23,27]. *Cronobacter* spp. have been isolated from a range of plant-origin foods, including ready-to-eat vegetables, cereals, and nuts [19,23,28]. Productive sources of *Cronobacter* strains are fresh or dried herbs and spices [29]. Although the bacterium has been isolated from various plant-based food products, no foodborne infections have been reported. The majority of *Cronobacter* infections occur in the adult population, but are less severe. Cases of *Cronobacter* infection in all age groups are probably under-reported for a number of reasons, such as misidentification as *Enterobacter cloacae* [29]. To date, very little is known about the genotyping diversity and hemolytic activity possessed by such plant-origin *Cronobacter*. The characterization of strains from plant-based foods with genomic features similar to clinically relevant strains of different *Cronobacter* spp. sequence types (STs) suggests that these foods may serve as potential vehicles for the transmission of opportunistic pathogens. Therefore, it is necessary to understand the genotyping diversity of *Cronobacter* spp. associated with plant-based foods to improve food safety.

As an intestinal pathogen, the main route of *Cronobacter* entry into the human body is through the consumption of contaminated food. These pathogens show a high tolerance to stressful environments, being one of the most heat-tolerant members of the *Enterobacteriaceae* family [5], extremely resistant to low water activity of the environment [30,31], able to tolerate acidic conditions as low as pH 4.2 [7] and produce biofilms that increase their survival under food production conditions [32]. After entering the host, the infection strategy consists of the following stages: (i) the colonization of the mucosa (intestinal, respiratory, or the urinary tract epithelia); (ii) the circumvention, subversion, and exploitation of host defenses; (iii) systemic spread and multiplication (within the blood or phagocytes); and (iv) host damage (through the expression of toxins and/or damage due to the proinflammatory modulation of the human immune system) [8].

Various hemolysins and hemolysis-related genes have been reported in *Cronobacter* genomes by Joseph et al. [4]. The hemolysin gene (*hly*) was present in all the genomes, with the only exceptions being *C. sakazakii* strain 701 and *C. malonaticus* strain 507. Most of the strains had two copies of the hemolysin gene and the hemolysin activator protein precursor gene [4]. Cruz et al. [33] identified the hemolysin gene (*hly*) as a hemolysin III homolog (COG1272), and Jang et al. [34], studying 390 strains, showed that all seven species tested possessed the hemolysin III COG1272 gene homolog. Additionally, three other hemolysin genes were identified, including genes encoding the cystathionine β-synthase (CBS) domain containing hemolysin, putative hemolysin, and 21 kDa hemolysin [34]. Umeda et al. [35] reported that all analyzed *Cronobacter* strains exhibited β-hemolytic activity against guinea pig, horse, and rabbit erythrocytes and that 92.9% of the strains were capable of the α-hemolysis of sheep erythrocytes. The further characterization of individual *Cronobacter* species, including strains occurring in food, is, however, needed to establish their hemolytic capacity. Results related to the hemolytic activity of *Cronobacter* spp. could extend the knowledge in this field, enabling the assessment of whether other closely related species commonly misidentified as *Cronobacter* (like *Franconibacter helveticus*, *Franconibacter pulveris*, *Siccibacter colletis*, and *Siccibacter turicensis*) differ in their hemolytic phenotypes from the *Cronobacter* genus species [36].

Multi-locus sequence typing (MLST) is a genetic method recommended for the characterization, differentiation, and typing of many microorganisms. Conventional MLST genotypes bacterial strains according to loci (commonly ~500 nucleotides) for seven housekeeping genes, and it enables the recognition of the sequence types (ST) and clonal complexes (CC) of the tested *Cronobacter* strains and thus the assessment of their genetic diversity and clinical significance [4,37].

The genetic basis of the virulence of *Cronobacter* spp. strains has not been fully elucidated, but some sequence types have been found to be associated with specific types of infection [37]. Infant meningitis due to *Cronobacter* spp. is more frequently caused by strains of *C. sakazakii* belonging to the CC4 clonal complex (especially ST4), whereas *C. sakazakii* ST12 is strongly associated with cases of necrotizing enterocolitis in infants. Infections in children and adults are mainly caused by the clonal complexes of *C. sakazakii* CC4 and *C. malonaticus* CC7, respectively [2,37].

Since the severity of infections due to *Cronobacter* spp. may be related to the genotype of the strain that caused the infection, it is important to determine the STs of isolates from both clinical and food sources. Therefore, the present study aimed to identify the sequence types of *Cronobacter* spp. strains isolated from commercial ready-to-eat leaf vegetables, sprouts, nuts, and dried fruits. The genotypic analysis of the strains could prove useful not only in epidemiological investigations, but also in their risk assessments.

## 2. Materials and Methods

### 2.1. Materials

The study was conducted with 24 strains of *Cronobacter* spp., including 15 strains of *C. sakazakii* (9n, 10n, 11m, s12, s14, s21, s22, s41, s42, s44, s45, s47, s48, lv25, and lv27), 6 *C. malonaticus* (5n, 6n, 7n, 8n, 12m, and lv31), 2 *C. turicensis* (1n and lv54), and 1 *C. condimenti* (s37). These isolates were from the bacterial collection of the Division of Milk Technology, Warsaw University of Life Sciences, Poland. The isolation of these strains from plant-derived food products, including ready-to-eat leaf vegetables, sprouts, nuts, and dried fruits, was described in our previous studies (Table 1) [19,23].

The *Cronobacter* spp. strains were stored frozen on Tryptone Soy Broth (TSB) (Oxoid Argenta, Poznań, Poland) with a 10% glycerol addition at a temperature of −40 °C. They were recovered from the frozen state by transferring 0.1 mL of a defrosted culture onto TSB of a given strain to 10 mL of sterile TSB, with incubation at a temperature of 35 °C for 24 h. Afterwards, each strain was inoculated onto Tryptone Soy Agar (TSA) medium (Oxoid Argenta, Poznań, Poland).

### 2.2. MLST Analysis

The MLST typing followed the methodology of Baldwin et al. [38]. The seven loci analyzed were glutaminyl tRNA synthetase gene (*glnS*), glutamate synthase large subunit gene (*gltB*), ATP synthase beta chain (*atpD*), DNA gyrase beta subunit (*gyrB*), phosphoenolpyruvate synthase A (*ppsA*), the gene encoding the translation initiation factor IF-2 (*infB*), and the gene encoding the translocase protein of the elongation factor EF-G (*fusA*). The gene fragments were amplified using primers and PCR conditions according to the protocol available in the *Cronobacter* MLST database https://pubmlst.org/organisms/cronobacter-spp/primers (accessed on 25 May 2020). The genomic DNA was isolated using the GenElute Bacterial Genomic DNA Kit (Sigma Aldrich, Poznań, Poland), according to the manufacturer’s instructions. The PCR and sequencing primers were synthesized at Eurofins Genomics (Ebersberg, Germany). The PCR was performed using the Phusion High-Fidelity PCR Master Mix with the HF buffer kit (ThermoFisher Scientific, Poland), in a total volume of 50 µL containing 20–40 ng of template DNA and 10 pmol of each primer. The polymerase used in the study was characterized by a 50-fold lower error insertion frequency compared to the standard Taq polymerase. The amplification products were purified using the GenElute PCR Clean-Up Kit (Sigma-Aldrich) or EXOSAP (ThermoFisher Scientific). The purified amplicons, obtained using standard and alternative primers (https://pubmlst.org/organisms/cronobacter-spp/primers and https://pubmlst.org/static/organisms/cronobacter-spp/Cronobacter_alternative_primers.pdf) (accessed on 20 June 2020), were sequenced in both directions (Eurofins Genomics). Afterwards, the obtained nucleotide sequences were compared with the sequences deposited in the *Cronobacter* MLST database (https://pubmlst.org/cronobacter/) (accessed on 17 October 2020). Alleles were determined for the seven loci, which identified their sequence type and clonal complex. Novel alleles and sequence types were assigned by the MLST database curator, Prof. Stephen Forsythe.

### 2.3. Hemolytic activity of Cronobacter spp. Strains

Hemolytic capability was determined on Columbia agar plates with a 5% addition of horse blood (COH) or sheep blood (COS) (Biomerieux, Warsaw Poland). Various hemolytic types were identified using the following reference strains: *Streptococcus pneumoniae* ATCC 6305, *Bacillus cereus* ATCC 14579, and *Listeria innocua* ATCC 33090 (Oxoid Argenta, Poznań Poland).

Single colonies of *Cronobacter* bacteria grown on TSA medium were transferred using a sterile, disposable loop to a blood agar plates and incubated at 37 °C. The hemolysis zones on the plates were measured after 96 h of incubation. The appearance of a green zone around the colony was recorded as α-hemolysis, a transparent zone around the colony was recorded as β-hemolysis, and no changes on the plate was defined as γ-hemolysis [39].

## 3. Results and Discussion

### 3.1. MLST Analysis

The MLST analysis results confirmed the earlier identification of all the tested strains obtained using RFLP-PCR: 15 strains belonged to the *C. sakazakii* species, 6 strains to the *C. malonaticus* species, 2 strains to the *C. turicensis* species, and one strain to the *C. condimenti* species. Interestingly, this profiling was consistent with the results of earlier intraspecies strain differentiation conducted with the RAPD-PCR method, indicating that the strains sharing the same RAPD pattern concurred with the same sequence type [19,23].

The present study showed that the 24 strains of *Cronobacter* spp. isolated from plant-derived foods (ready-to-eat leaf vegetables, sprouts, nuts, and dried fruits) belonged to 14 different sequence types (ST) (Table 2).

The main STs identified were *C. sakazakii* ST99 (40%, 6/15), which was isolated from various sprouts, and *C. malonaticus* ST258 (66.7%, 4/6), which was isolated from various nuts. Interestingly, out of the 24 strains, six novel sequence types were defined: ST806 and ST808 in the *C. turicensis* 1n and lv54 strains; ST805 and ST807 in the *C. malonaticus* 12m and lv31 strains; and ST804 in the *C. sakazakii* s41 and s42 strains. In the case of the *C. malonaticus* 12m strain, the analyses showed the presence of a novel allele in the *ppsA* locus and the closest match to the 355 allele with four differences (A231G, G291A, T387G, and T438C). The *C. turicensis* lv54 strain showed novel alleles in as many as four loci, i.e., *gltB*, *gyrB*, *infB*, and *ppsA*. The sequencing chromatograms indicates as the closest match *pps*159 (C48T, C57T, C405G, and T459C), *gltB*64 (A252G, T351C, and T480C), *infB*80 (C21G), and *gyrB*31 (C96T and T138C). The remaining detected sequence types included: ST4, ST17, ST648, ST21, ST494 (*C. sakazakii*), and ST98 (*C. condimenti*). Among the 15 *C. sakazakii* isolates, 4 (s12, 9n, 10m, and 11m,) had STs: ST4, ST17, and ST494, assigned to isolates from clinical sources in the PubMLST *Cronobacter* database (accessed: 10 May 2023).

It is notable that *C. sakazakii* strains belonging to the sequence type ST4 and the clonal complex CC4 are often associated with severe infections leading to meningitis in neonates, children, and adults. These strains have been isolated from milk powder, food products, ice creams, and powdered infant formulas [2,37]. In the present study, the only ST4 strain was *C. sakazakii* s12 (6.6%, 1/15), which was isolated from alfalfa sprouts. Wang et al. [40] analyzed 84 *Cronobacter* spp. isolates obtained from foods imported to Beijing in 2006–2015 and identified mainly the pathogenic sequence types ST1 and ST4 in the case of 31.67% (19/60) and 21.67% (13/60) of *C. sakazakii* strains, respectively, as well as the ST7 type in 70.59% (12/17) of *C. malonaticus* strains. The sequence type ST1 belongs to the clonal complex 1 of *C. sakazakii* (CC1), which is the second major ST in the PubMLST database after ST4 (CC4). ST4 (*C. sakazakii*) and ST7 (*C. malonaticus*) were the predominant STs identified by Li et al. [41] for the *Cronobacter* spp. isolated from wet rice and flour products in China. Similarly, Fei et al. [42] determined the sequence types ST4 (19/56, 33.9%), ST1 (12/56, 21.4%), and ST64 (11/56, 16.1%) in *C. sakazakii* as dominant for *Cronobacter* spp. isolated from powdered infant formula collected from Chinese retail markets. The results of this study may provide a theoretical basis for investigating the transmission routes and genotyping diversity of *Cronobacter* spp. and developing more effective methods for preventing this organism.

### 3.2. Hemolytic Activity of Cronobacter spp.

The hemolytic activity of the *Cronobacter* strains was determined (Table 3). In total, 87.5% and 16.7% of the *Cronobacter* spp. strains were capable of β-hemolysis on the culture medium with the addition of horse and sheep red blood cells, respectively.

A greater capability for *β*-hemolysis was determined on the horse blood agar, as indicated by a zone width ranging from 1.0 to 3.5 mm, than on the sheep blood agar—a zone width ranging from 1.0 to 1.2 mm. Regardless of ST and CC, all the *C. sakazakii* strains produced β-hemolysis on the horse blood agar and 86.7% of the strains caused α-hemolysis on the sheep blood agar. Both *C. sakazakii* strains (s41 and s42) belonging to ST 804 were capable of causing β–hemolysis on both blood agars. In turn, both analyzed *C. turicensis* (1n and lv54) strains assigned to different STs (806 and 808) were α-hemolytic on the horse blood agar and weakly β-hemolytic on the sheep blood agar. All the analyzed strains of *C. malonaticus* were capable of α-hemolysis on the sheep blood agar, while one strain assigned to ST 807 was α-hemolytic on both blood agar media. In the case of *C. condimenti* s37, β-hemolysis was shown on the horse blood agar, whereas it caused α-hemolysis on the sheep blood agar. All the *Cronobacter* strains assigned to a given ST or CC showed the same hemolytic activity profile on both blood agar media. The obtained results of the hemolytic activity of the *Cronobacter* strains indicated that the type of hemolysis was not species-specific, but was a strain-dependent feature.

The presence of various hemolysins and hemolysin-related genes in *Cronobacter* spp. has been described by many authors [33,34,43]. Fakruddin et al. [44] showed hemolytic activity on human blood agar with two out of six *C. sakazakii* strains isolated from food samples, whereas Rajani et al. [45] demonstrated β-hemolysis on bovine blood agar after a 4-day incubation at a temperature of 37 °C in the case of all 11 *C. sakazakii* isolates tested. Furthermore, Cui et al. [46] found that 27 of 31 (87%) *Cronobacter* isolates were not capable of hemolysis, except for *C. sakazakii* SC26, *C. malonaticus* SD16, *C. malonaticus* SD26, and *C. muytjensii* SD83, as indicated by the cleared zones produced around them. Finally, Umeda et al. [35] reported that 57 (100%) *Cronobacter* strains exhibited β-hemolytic activity against guinea pig, horse, and rabbit erythrocytes, and that 92,9% of the strains were capable of the α-hemolysis of sheep erythrocytes. It is known that a gene encoding a hemolysin is present in *Cronobacter* spp. [33], however, whether this hemolysin is active and associated with cytotoxicity has not yet been clarified. This requires conducting more in-depth genetic studies to assign the functionality of these various hemolysin genes to the appropriate phenotype.

## 4. Conclusions

The present study demonstrated the considerable genotypic diversity of *Cronobacter* strains isolated from plant-based ready-to-eat foods. Five novel sequence types (804, 805, 806, 807, and 808) were detected and novel alleles were found in four loci: *ppsA, gltB, gyrB*, and *infB*. Therefore, future studies should aim to detect more new STs, which, in the study of the genotypic diversity of *Cronobacter* spp., are beneficial for monitoring some sources and will enable the development of more effective methods to control these bacteria. The sequence type of one *C. sakazakii* strain (s12) was found to be ST4, which is strongly associated with meningitis in newborns. It was also found that 87.5% of the *Cronobacter* spp. strains were capable of β-hemolysis on the culture medium with horse red blood cells and 16.7% on the medium with sheep red blood cells. The presence of the pathogenic species *C. sakazakii*, *C. malonaticus*, and *C. turicensis* in ready-to-eat plant-derived food products shows they are potential sources of infection, especially to those with compromised immunity, which substantiates their further multi-faceted characterization [47]. The fact that active hemolysins appear in isolates from plant-based food seems to be disturbing, as this feature is usually associated with pathogens originating from clinical sources.

## Figures and Tables

**Table 1 foods-12-03873-t001:** Strains of *Cronobacter* spp. used in the study and their origin.

Isolate	Species	Origin
s12	*C. sakazakii*	Alfalfa sprouts
9n	*C. sakazakii*	Brazilian nuts
10n	*C. sakazakii*	Brazilian nuts
14	*C. sakazakii*	Alfalfa sprouts
s44	*C. sakazakii*	Mix of sprouts
s45	*C. sakazakii*	Mix of sprouts
s21	*C. sakazakii*	Leek sprouts
s47	*C. sakazakii*	Mix of sprouts
s48	*C. sakazakii*	Mix of sprouts
s22	*C. sakazakii*	Leek sprouts
11m	*C. sakazakii*	Mixes of dried fruits, seeds, and nuts
lv25	*C. sakazakii*	Rucola
lv27	*C. sakazakii*	Endive escarola
s41	*C. sakazakii*	Sunflower sprouts
s42	*C. sakazakii*	Sunflower sprouts
5n	*C. malonaticus*	Hazelnuts
6n	*C. malonaticus*	Cashew nuts
7n	*C. malonaticus*	Pini nuts
8n	*C. malonaticus*	Macadamia nuts
12m	*C. malonaticus*	Mixes of dried fruits, seeds, and nuts
lv31	*C. malonaticus*	Lambs lettuce
1n	*C. turicensis*	Almonds
lv54	*C. turicensis*	Mix of leaf vegetables
s37	*C. condimenti*	Small radish sprouts

**Table 2 foods-12-03873-t002:** Comparison of 7-loci MLST for 24 strains from 4 *Cronobacter* species.

PubMLST ID	Isolate	Species	*atpD*	*fusA*	*glnS*	*gltB*	*gyrB*	*infB*	*ppsA*	ST	CC
4062	s12	*C. sakazakii*	5	1	3	3	5	5	4	4	4
4063	9n	*C. sakazakii*	3	12	16	5	16	20	14	17	17
4064	10n	*C. sakazakii*	3	12	16	5	16	20	14	17	17
4065	s14	*C. sakazakii*	3	11	13	18	11	17	13	21	21
4066	s44	*C. sakazakii*	3	8	52	54	21	65	73	99	99
4067	s45	*C. sakazakii*	3	8	52	54	21	65	73	99	99
4068	s21	*C. sakazakii*	3	8	52	54	21	65	73	99	99
4069	s47	*C. sakazakii*	3	8	52	54	21	65	73	99	99
4070	s48	*C. sakazakii*	3	8	52	54	21	65	73	99	99
4071	s22	*C. sakazakii*	3	8	52	54	21	65	73	99	99
4072	11m	*C. sakazakii*	11	8	24	220	15	56	261	494	-
4073	lv25	*C. sakazakii*	175	1	120	275	21	234	333	648	-
4074	lv27	*C. sakazakii*	175	1	120	275	21	234	333	648	-
3574	s41	*C. sakazakii*	3	1	120	94	270	1	368	** 804 **	-
3575	s42	*C. sakazakii*	3	1	120	94	270	1	368	** 804 **	-
4075	5n	*C. malonaticus*	89	13	107	8	10	35	160	258	-
4076	6n	*C. malonaticus*	89	13	107	8	10	35	160	258	-
4077	7n	*C. malonaticus*	89	13	107	8	10	35	160	258	-
4078	8n	*C. malonaticus*	89	13	107	8	10	35	160	258	-
3139	12m	*C. malonaticus*	64	7	64	7	10	16	** 381 **	** 805 **	-
3576	lv31	*C. malonaticus*	3	8	10	94	5	93	74	** 807 **	-
3573	1n	*C. turicensis*	46	147	42	21	237	193	318	** 806 **	-
3140	lv54	*C. turicensis*	46	5	4	** 314 **	** 279 **	** 265 **	** 382 **	** 808 **	-
1896	s37	*C. condimenti*	24	86	96	28	63	42	147	98	-

New assigned allele number and STs are underlined and in bold.

**Table 3 foods-12-03873-t003:** Hemolysis of horse blood and sheep blood agars by *Cronobacter* spp. strains.

Isolate	Species	Hemolysis (Zone in mm)	
		Horse Blood Agar	Sheep Blood Agar
s12	*C. sakazakii*	β (2.9)	α
9n	*C. sakazakii*	β (2.4)	α
10n	*C. sakazakii*	β (2.8)	α
s14	*C. sakazakii*	β (2.7)	α
s44	*C. sakazakii*	β (3.3)	α
s45	*C. sakazakii*	β (2.8)	α
s21	*C. sakazakii*	β (2.4)	α
s47	*C. sakazakii*	β (2.5)	α
s48	*C. sakazakii*	β (2.8)	α
s22	*C. sakazakii*	β (3.2)	α
11m	*C. sakazakii*	β (1.0)	α
lv25	*C. sakazakii*	β (2.5)	α
lv27	*C. sakazakii*	β (1.1)	α
s41	*C. sakazakii*	β (3.5)	β (1.2)
s42	*C. sakazakii*	β (3.1)	β (1.0)
5n	*C. malonaticus*	β (1.9)	α
6n	*C. malonaticus*	β (1.0)	α
7n	*C. malonaticus*	β (2.6)	α
8n	*C. malonaticus*	β (2.4)	α
12m	*C. malonaticus*	β (1.0)	α
lv31	*C. malonaticus*	α	α
1n	*C. turicensis*	α	β (1.0)
lv54	*C. turicensis*	α	β (1.0)
s37	*C. condimenti*	β (1.0)	α

## Data Availability

Data are contained within the article.

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
