# Peer review of "The Genotyping Diversity and Hemolytic Activity of Cronobacter spp. Isolated from Plant-Based Food Products in Poland"

_foods, 2023, doi:10.3390/foods12203873_

Round 1

Reviewer 1 Report

Comments and Suggestions for Authors

The manuscript is a good one. I expected the authors to analyse more than 24 isolates and profusely discuss the public health implications of finding zoonotic and haemolytic species of Cronobacter in foods.

Specific comments for authors

  1. Introduction: This section should provide sufficient background on how Cronobacter spp contributes to human health problems due to the consumption of foods, especially  commercial ready-to-eat leaf vegetables, sprouts, nuts, and dried fruits 
  1. Lines 69-80: Were all the genes harboured by Cronobacter species isolated from foods or are they from clinical isolates? 
  1. Lines 87-89: Citation(s) required 
  1. Line 107: I feel that the 24 isolates are small for generalization concerning the Genetic diversity and hemolytic activity of Cronobacter spp. isolated from plant-based food products in Poland. Are the 24 isolates all you could find/isolate? 
  1. Table 1: Would the authors like to determine if there is a statistical association between Cronobacter spp. isolation and food types (origin)?
  2. Line 157:  Why did the authors incubate at 35°C instead of 37°C? 
  1. Lines 156-159:  Citation(s) required 
  1. Discussion: I expect the authors to discuss the public health implications of finding zoonotic and hemolytic species of Cronobacter in foods and also suggest the way out.

Author Response

Thank you very much for reviewing our manuscript: Genotyping diversity and hemolytic activity of Cronobacter spp. isolated from plant-based food products in Poland. We have adopted all your suggestions.

Your suggestions have seriously contributed to the improvement of our manuscript. All changes compared to the original version have been highlighted in yellow. Hope the revised manuscript will be evaluated as improved, in any case, we are willing to consider any further request.

  1. Introduction: This section should provide sufficient background on how Cronobacter spp. contributes to human health problems due to the consumption of foods, especially commercial ready-to-eat leaf vegetables, sprouts, nuts, and dried fruits

Introduction was revised.

  1. Lines 69-80: Were all the genes harboured by Cronobacter species isolated from foods or are they from clinical isolates?

Hemolysis-related genes have been reported in Cronobacter genomes isolated from foods, clinical specimens, from human and nonhuman sources, adult filth flies.

  1. Lines 87-89: Citation(s) required

Citation was added.

  1. Line 107: I feel that the 24 isolates are small for generalization concerning the Genetic diversity and hemolytic activity of Cronobacter spp. isolated from plant-based food products in Poland. Are the 24 isolates all you could find/isolate?

The 24 isolates tested in this study are not all strains that we isolated from plant-based food.  For this study were selected three Cronobacter species being of key significance from the clinical perspective, i.e.: C. sakazakii, C. malonatiocus, C. turicensis. Additionally, due to the fact that the C. condimentii s37 strain we isolated is the second strain in the world, it was also included in this study.

Table 1: Would the authors like to determine if there is a statistical association between Cronobacter spp. isolation and food types (origin)?

This is a very valuable comment and a good tip for us to plan further research. After completing the research part that we are currently conducting, we will definitely take into account the statistical association between this two factors.

  1. Line 157:  Why did the authors incubate at 35°C instead of 37°C?

            Thank you for finding the mistake. The plates were incubated at 37°C according to the method: Buxton R. Blood Agar Plates and Hemolysis Protocols. American Society for Microbiology, 2005, 1-9. https://asm.org/getattachment/7ec0de2b-bb16-4f6e-ba07-2aea25a43e76/protocol-2885.pdf.We corrected it in the manuscript.

  1. Lines 156-159:  Citation(s) required

The methodology for determining hemolytic activity has been added.

  1. Discussion: I expect the authors to discuss the public health implications of finding zoonotic and hemolytic species of Cronobacter in foods and also suggest the way out.

Revised in modified manuscript.

Reviewer 2 Report

Comments and Suggestions for Authors

The manuscript entitled “Genotyping diversity and hemolytic activity of Cronobacter spp. isolated from plant-based food products in Poland” describes genotyping diversity and hemolytic properties of 24 strains of Cronobacter spp. isolated from commercial ready-to-eat leaf vegetables, sprouts, nuts, and dried fruits (isolated and reported in a previous study). The article reports a short data and following concerns need to be addressed

Ø  The authors determined the sequence types (ST) and clonal complexes (CC) of Cronobacter spp. isolated from a single type of food matrix, namely, plant-based foods. Typically, diversity studies are most informative when the isolates are sourced from multiple origins. For example, including isolates from food samples, human clinical samples, livestock, and environmental sources. This raises questions about the potential insights’ readers can derive from this article.

Ø  The authors conducted MLST analysis in their study. Nowadays, whole genome sequencing (WGS) analysis is commonly employed to gain insights into the genomic epidemiology of foodborne pathogens. The choice of MLST analysis by the authors raises the question of why they opted for this method instead. Explain and describe the rationale and mention the same in limitations.

Ø  Highlight the clinical significance of C. sakazakii, C. malonaticus, and C. turicensis, and provide an overview of previous reports documenting foodborne outbreaks linked to ready-to-eat plant-based foods caused by these pathogens.

Ø  Lines 88-89: “Conventional MLST genotypes according to loci for seven housekeeping genes” The sentence is incomplete.

Ø  Lines 89-91: Rephrase, again incomplete

Ø  Line 133: Eurofins Genomics, Mention the firm name, city and country name

Ø  Hemolytic activity of Cronobacter spp. strains- reference, Any standard method followed?

Ø  Lines 151-153: Various hemolytic types were identified using the following reference strains: Streptococcus pneumoniae ATCC 6305, Bacillus cereus ATCC 14579, and Listeria innocua ATCC 33090- What does it mean?

Ø  Line 155: TSA culture?

Ø  Cronobacter spp. is not known for causing hemolysis, and the classification of hemolysis types (alpha, beta, gamma) is more commonly used for other bacteria, particularly some strains of Streptococcus and Staphylococcus. Therefore, for Cronobacter, you would not typically use terms like alpha, beta, or gamma hemolysis to describe its characteristics. Instead, you would focus on other aspects of its biology and virulence factors when studying this bacterium.

Ø  he authors' discussion of the results is limited. I strongly recommend that the authors rewrite the discussion section, incorporating references to and comparisons with previously conducted MLST analyses in Cronobacter spp. Additionally, they should elaborate on novel sequence types (STs) and their potential implications in human diseases.

Ø  Conclusion- Rewrite. Focus on novel STs

Comments on the Quality of English Language

Extensive editing of English language required

Author Response

Thank you very much for reviewing our manuscript: Genotyping diversity and hemolytic activity of Cronobacter spp. isolated from plant-based food products in Poland. We have adopted all your suggestions.

Your suggestions have seriously contributed to the improvement of our manuscript. All changes compared to the original version have been highlighted in yellow. Hope the revised manuscript will be evaluated as improved, in any case, we are willing to consider any further request.

Ø  The authors determined the sequence types (ST) and clonal complexes (CC) of Cronobacter spp. isolated from a single type of food matrix, namely, plant-based foods. Typically, diversity studies are most informative when the isolates are sourced from multiple origins. For example, including isolates from food samples, human clinical samples, livestock, and environmental sources. This raises questions about the potential insights’ readers can derive from this article.

Due to the fact that Cronobacter occurs commonly in food, it is important to characterize strains isolated from various sources. In our research, we focused on material derived from plant products because, compared to, for example, research on isolates derived from infant formulas, there is still little research in this field. However, it would obviously be valuable to characterize and compare strains from various food sources and strains from clinical sources.

Ø  The authors conducted MLST analysis in their study. Nowadays, whole genome sequencing (WGS) analysis is commonly employed to gain insights into the genomic epidemiology of foodborne pathogens. The choice of MLST analysis by the authors raises the question of why they opted for this method instead. Explain and describe the rationale and mention the same in limitations.

The multi-locus sequence typing (MLST) is a genetic method recommended for the characterization, differentiation and typing of many microorganisms. Conventional MLST genotypes bacterial strains according to loci (commonly ~500 nucleotides) for seven housekeeping genes, it enables the recognition of sequence types (ST) and clonal complexes (CC) of the tested Cronobacter strains, and thus the assessment of their genetic diversity and clinical significance (which we it depended).

Of course, we agree that whole genome sequencing (WGS) analysis is now widely used to gain insight into the epidemiology of the genome, but for genotypic diversity and determination of ST and CC in Cronobacter, the MLST method seems to be sufficient.

Ø  Highlight the clinical significance of C. sakazakii, C. malonaticus, and C. turicensis, and provide an overview of previous reports documenting foodborne outbreaks linked to ready-to-eat plant-based foods caused by these pathogens.

The clinical significance of C. sakazakii, C. malonaticus, and C. turicensis was highlighted.

We have not found any data in the available scientific literature on food poisoning related to the consumption of ready-to-eat food contaminated with Cronobacter.

Ø  Lines 88-89: “Conventional MLST genotypes according to loci for seven housekeeping genes” The sentence is incomplete.

Revised in modified manuscript

Ø  Lines 89-91: Rephrase, again incomplete

Revised in modified manuscript

Ø  Line 133: Eurofins Genomics, Mention the firm name, city and country name

Revised in modified manuscript

Ø  Hemolytic activity of Cronobacter spp. strains- reference, Any standard method followed?

Method has been added.

Ø  Lines 151-153: Various hemolytic types were identified using the following reference strains: Streptococcus pneumoniae ATCC 6305, Bacillus cereus ATCC 14579, and Listeria innocua ATCC 33090- What does it mean?

The reference strains show different hemolysis and were used as positive controls: Streptococcus pneumoniae ATCC 6305 - for α-hemolysis, Bacillus cereus ATCC 14579 - for β-hemolysis and Listeria innocua ATCC 33090 - for γ-hemolysis.

Ø  Line 155: TSA culture?

The sentence has been corrected.

Ø  Cronobacter spp. is not known for causing hemolysis, and the classification of hemolysis types (alpha, beta, gamma) is more commonly used for other bacteria, particularly some strains of Streptococcus and Staphylococcus. Therefore, for Cronobacter, you would not typically use terms like alpha, beta, or gamma hemolysis to describe its characteristics. Instead, you would focus on other aspects of its biology and virulence factors when studying this bacterium.

The obtained results of tests on the hemolytic activity of Cronobacter strains showed that they are capable of causing various types of hemolysis (see Table 3). We agree that Cronobacter has multiple virulence factors and understanding them is very important. Our goal was not to understand the virulence of Cronobacter, but to better understand their hemolytic characteristics, which are poorly known. Specific types of hemolysis are defined for many pathogenic microorganisms. A routine medium for microbiological diagnostics is blood agar, so the type of hemolysis may be an important feature enabling their virulence, as in other bacteria. The literature data cited in the manuscript [Fakruddin et al., Rajani et al., Cui et al., Umeda et al.] indicate that Cronobacter causes hemolysis, but determining this feature is omitted by most researchers. That's why it was our goal. We believe that hemolytic activity may also be important in differentiating Cronobacter from closely related species often mistakenly identified as Cronobacter.

Ø  The authors' discussion of the results is limited. I strongly recommend that the authors rewrite the discussion section, incorporating references to and comparisons with previously conducted MLST analyses in Cronobacter spp. Additionally, they should elaborate on novel sequence types (STs) and their potential implications in human diseases.

Revised in modified manuscript

Ø  Conclusion- Rewrite. Focus on novel STs

Revised in modified manuscript

Reviewer 3 Report

Comments and Suggestions for Authors

In this study, the authors reported that “Genotyping diversity and hemolytic activity of Cronobacter spp. isolated from plant-based food products in Poland”. The study indicates that the author used a total of 24 Cronobacter strains, which were isolated early and reported. In this study, the authors investigated their genotyping and hemolytic activity. However, this study's results are not enough to recommend this study for publication in foods.

The abstract is not written well. The author should deliver the abstract with the aim, significance, objectives, and primary results of this study in a constructive manner.

Why Cronobacter species was mainly chosen in this study? The environmental condition of ready-to-eat leaf vegetables should be given.

The author must present the hemolytic activity experimental data.

Comments on the Quality of English Language

 Minor editing of English language required

Author Response

Thank you very much for reviewing our manuscript: Genotyping diversity and hemolytic activity of Cronobacter spp. isolated from plant-based food products in Poland. We have adopted all your suggestions.

Your suggestions have seriously contributed to the improvement of our manuscript. All changes compared to the original version have been highlighted in yellow. Hope the revised manuscript will be evaluated as improved, in any case, we are willing to consider any further request.

In this study, the authors reported that “Genotyping diversity and hemolytic activity of Cronobacter spp. isolated from plant-based food products in Poland”. The study indicates that the author used a total of 24 Cronobacter strains, which were isolated early and reported. In this study, the authors investigated their genotyping and hemolytic activity. However, this study's results are not enough to recommend this study for publication in foods.

  1. The abstract is not written well. The author should deliver the abstract with the aim, significance, objectives, and primary results of this study in a constructive manner.

The aim of this study was pointed in the first sentence (higligted in yellow).

Abstract was rewriten according to the reviewer sugestion.

  1. Why Cronobacter species was mainly chosen in this study? The environmental condition of ready-to-eat leaf vegetables should be given.

Cronobacter has long been within the scope of our scientific research. So far, we have managed to isolate strains that have not been described in the literature and which may cause food poisoning. The general microflora of ready-to-eat leaf vegetables has been characterized by us previously (Berthold-Pluta, A.; Garbowska, M.; StefaÅ„ska, I.; Pluta, A. Microbiological quality of selected ready-to-eat leaf vegetables, sprouts and non-pasteurized fresh fruit-vegetable juices including the presence of Cronobacter spp. Food Microbiol. 2017, 65, 221–230. http://doi.org/10.1016/j.fm.2017.03.005, 23.            Berthold-Pluta, A.; Garbowska, M.; StefaÅ„ska, I.; Stasiak-RóżaÅ„ska, L.; Aleksandrzak-Piekarczyk, T.; Pluta, A. Microbiologi-cal quality of nuts, dried and candied fruits, including the prevalence of Cronobacter spp. Pathogens 2021, 10(7), 1–13. http://doi.org/10.3390/pathogens10070900). Literature data on the genotypic characteristics of Cronobacter strains originating from ready to eat food are limited.

In the reviewed manuscript, the research material included isolates obtained in previous studies.

  1. The author must present the hemolytic activity experimental data.

The hemolytic activity of the strains was determined by hemolysis zones according to Buxton R. Blood Agar Plates and Hemolysis Protocols. American Society for Microbiology, 2005, 1-9. https://asm.org/getattachment/7ec0de2b-bb16-4f6e-ba07-2aea25a43e76/protocol-2885.pdf, and the results are summarized in Table 3.

Round 2

Reviewer 2 Report

Comments and Suggestions for Authors

Comments addressed

Comments on the Quality of English Language

Minor editing of the English language required

Reviewer 3 Report

Comments and Suggestions for Authors

-